# Corticoreticular Pathway in Post-Stroke Spasticity: A Diffusion Tensor Imaging Study

**DOI:** 10.3390/jpm11111151

**Published:** 2021-11-04

**Authors:** Sung-Hwa Ko, Taehyung Kim, Ji Hong Min, Musu Kim, Hyun-Yoon Ko, Yong-Il Shin

**Affiliations:** 1Department of Rehabilitation Medicine, Pusan National University Yangsan Hospital, Yangsan 50612, Korea; ijsh6679@gmail.com (S.-H.K.); papered@hanmail.net (J.H.M.); dr.mskim90@gmail.com (M.K.); drkohy@gmail.com (H.-Y.K.); 2Research Institute for Convergence of Biomedical Science and Technology, Pusan National University Yangsan Hospital, Yangsan 50612, Korea; thkim1981@gmail.com; 3Department of Rehabilitation Medicine, School of Medicine, Pusan National University, Yangsan 50612, Korea

**Keywords:** corticoreticular pathway, diffusion tensor imaging, reticulospinal tract, spasticity, stroke

## Abstract

One of the pathophysiologies of post-stroke spasticity (PSS) is the imbalance of the reticulospinal tract (RST) caused by injury to the corticoreticular pathway (CRP) after stroke. We investigated the relationship between injuries of the CRP and PSS using MR diffusion tensor imaging (DTI). The subjects were divided into spasticity and control groups. We measured the ipsilesional fractional anisotropy (iFA) and contralesional fractional anisotropy (cFA) values on the reticular formation (RF) of the CRP were on the DTI images. We carried out a retrospective analysis of 70 patients with ischemic stroke. The cFA values of CRP in the spasticity group were lower than those in the control group (*p* = 0.04). In the sub-ROI analysis of CRP, the iFA values of pontine RF were lower than the cFA values in both groups (*p* < 0.05). The cFA values of medullary RF in the spasticity group were lower than the iFA values within groups, and also lower than the cFA values in the control group (*p* < 0.05). This results showed the CRP injury and that imbalance of RST caused by CRP injury was associated with PSS. DTI analysis of CRP could provide imaging evidence for the pathophysiology of PSS.

## 1. Introduction

Stroke is a leading cause of long-term disability [1]. More than two-thirds of patients with stroke have various sequelae, of which motor impairment is the most common [2]. One of the common motor consequences of stroke is spasticity. Spasticity is a motor disorder characterized by a velocity-dependent increase in the muscle tone, resulting from hyperexcitability of the stretch reflexes as one component of the upper motor neuron syndrome [3].

Post-stroke spasticity (PSS) has been reported in 4–46% of patients with stroke, and the prevalence increases from the acute to chronic state after stroke [4]. The predictive factors of PSS are severe paresis, hypoesthesia, lower initial stroke severity, lower activity of living, and previous stroke [5]. However, the pathophysiologic mechanisms underlying PSS remain poorly understood. Some previous studies have suggested the role of the reticulospinal tract (RST) in PSS [6,7,8].

The RST consists of two components. The dorsal RST, which originates from the reticular formation (RF) in the medulla, provides inhibitory inputs to spinal reflex circuits, while the medial RST, which originates from the RF in the pons, provides excitatory inputs. The CRP arises from the premotor cortex (PMC) and supplementary motor area (SMA) and synapses with the neurons of the RST in RF [9]. The dorsal RST receives inputs primarily from the contralateral motor cortex, which descends ipsilaterally to the spinal cord, while the medial RST receives inputs primarily from the ipsilateral PMC/SMA, which descends ipsilaterally to the spinal cord [10]. If a stroke occurs in the cerebral hemisphere, the CRP from ipsilesional PMC reduces the inputs of the contralesional medullar RF and resulting in hypoactivity of the inhibitory effects of contralesional dorsal RST to the spinal stretch reflex. In addition, CRP from contralesional PMC to the contralesional pontine RF becomes unopposed and gradually hyperexcitable, which results in the hyperactivity of excitability effects of contralesional medial RST [6,10]. In other words, one of the pathophysiologies of PSS is the imbalance of dorsal RST and medial RST, and it is caused by injury to the CRP after stroke.

Neuroimaging examinations, such as functional MRI, diffusion tensor imaging (DTI), and functional near-infrared spectroscopy (fNIRS), are the main assessment tools to investigate post-stroke motor impairment, and the neurological biomarkers are correlated with motor function after stroke. Most neuroimaging studies of the stroke-related motor pathway assessed the corticospinal tract (CST) [11,12]. In particular, as the number of DTI studies on CST after stroke has increased and the evidence has become stronger, DTI biomarkers of CST have been suggested as predictors of motor recovery [13,14].

Recently, studies on identification of CRP in the healthy brain using DTI have been reported [15,16]. Since then, studies on CRP injury after stroke and traumatic brain injury have also been conducted using DTI. However, there have been few DTI studies of CRP after stroke. The DTI studies on CRP after stroke only referred to the relationship between CRP and gait function [17,18,19,20].

In this study, we attempted to investigate the relationship between CRP injury and spasticity in stroke patients by using DTI. Also, we analyzed CRP at the level of the RF in the brainstem, where CRP and each RST are contiguous; we want to confirm that the that PSS is caused by an imbalance between the dorsal RST and the medial RST due to CRP injury after stroke.

## 2. Materials and Methods

### 2.1. Study Design and Population

This study was a retrospective, single-center study. The study protocol was approved by the Institutional Review Board of Pusan National University of Yangsan Hospital. The requirement of informed consent was waived because the data were analyzed anonymously and retrospectively.

### 2.2. Study Population

Ischemic stroke patients admitted to our hospital between January 2015 and December 2019 were included. We identified ischemic stroke by brain MR. Subjects were recruited according to the following inclusion criteria: (1) first-ever stroke, (2) age between 18 and 80 years, (3) ischemic stroke confined to the supratentorial level, (4) hemiplegic stroke, (5) DTI performed at three to six weeks after the onset of stroke. Exclusion criteria were as follows: (1) transient ischemic attack or hemorrhagic stroke, (2) bi-hemispheric or brain stem lesions, (3) hemorrhagic transformation or cerebral/cerebellar edema after cerebral infarction, (4) coma state or no neurologic symptom after stroke, (5) any previous brain lesion such as previous stroke, traumatic brain injury, brain tumor, etc. (6) any previous anatomical or muscular abnormality of hemiplegic limbs. Patients enrolled through inclusion and exclusion criteria were divided into two groups according to the presence of spasticity. All subjects underwent conventional inpatient rehabilitation including physical therapy and occupational therapy after stroke in our center.

### 2.3. Data Collection and Variables

The data were obtained from the electronic chart review. The following data were retrieved: age, sex, side and lesion of stroke, severity of the stroke, interval between onset and MR DTI, modified Ashworth scale (MAS) score at discharge. 

### 2.4. Assessment of Spasticity

Spasticity was assessed based on the MAS score of the elbow flexor of the affected upper extremity at discharge. The MAS grades the resistance of a relaxed limb to rapid passive stretch through the range of motion by determining the score with 6-point scales while providing reliable and reproducible results [21,22]. An MAS score of 0 indicates no increase in muscle tone, and a score of 4 indicates a state in which passive movement of the affected limb is impossible. In this study, any spasticity was defined as an MAS score ≥1 for the elbow flexor that performed the passive movements.

### 2.5. MR Data Acquisition

All subjects underwent 3.0T MRI scanner (Skyra, Siemens Healthneers, Germany) equipped with a 16-channel head & neck coil to acquire 3D T1-weighted and DTI images. The 3D T1-weighted images were obtained using magnetization prepared rapid acquisition gradient recalled echo (MPRAGE) pulse sequence with the following parameters: repetition time (TR)/echo time (TE)/inversion time (TI) = 1900/2.2/900 ms, Flip angle (FA) = 9°, 1 mm^3^ isotropic voxel size. DTI data were acquired at an average of 32.1 ± 7.0 days after stroke onset by conventional brain MRI protocols using the echo-planar imaging sequence.

DTI was applied in diffusion-weighted gradients along 60 non-collinear directions and two volumes without diffusion weighting. The imaging parameters were matrix size = 128 × 128, field of view = 230 × 230 mm^2^, repetition time (TR) = 5800 ms, echo time = 71 ms, flip angle = 180°, number of averages = 1, b value = 1000 s/mm^2^, slice thickness = 4 mm, and voxel size = 1.7 × 1.7 × 4 mm^3^. To accelerate data acquisition, parallel imaging with an acceleration factor of two was applied.

### 2.6. DTI Data Processing

All DTI data was spatially normalized to the Montreal Neurological Institute (MNI) template using parameters derived from 3D T1WI processing. DTI analysis was performed with FMRIB’s Diffusion Toolbox and TBSS (Tract-Based Spatial Statistics) in the FMRIB Software Library 6.0 (FSL, https://fsl.fmrib.ox.ac.uk/fsl/fslwiki/FSL, accessed on: 2 June 2021) package. For registration, was performed with FMRIB Software Library 6.0 and get the transformation matrix with FMRIB’s linear registration tool (FLIRT) and FMRIB’s nonlinear registration tool(FNIRT). Next, the eddy current distortions were corrected in the DTI datasets. After this, raw diffusion-weighted images for each subject were linearly aligned to non-diffusion weighted image (b0), followed by removal of the non-brain tissues using a brain extraction tool (BET). Third, the extracted brain was used for local fitting of diffusion tensors. The diffusion tensor was calculated at each voxel to generate the fractional anisotropy (FA) images.

TBSS can be considered as the standard approach for voxel-based analysis (VBA) of diffusion tensor imaging (DTI) data. All individual FA images were linearly and nonlinearly aligned to a 1-mm isotropic FA template in standard MNI space using FNIRT. To create the mean FA skeleton that served as a study-specific template, all aligned FA images were averaged and thinned by local-perpendicular non-maximum suppression with FA thresholds of 0.2 to exclude the voxels in GM and cerebrospinal fluid (CSF). The resulting skeleton represented the center of common WM tracts. Fiber tracking was performed using a probabilistic tractography method.

QFA values were obtained by manually placing ROIs on the entire CRP and CST areas at the level of the lower pons to upper medulla on axial sections (left and right sides) on the basis of the T2-weighted image and anatomic knowledge, by using our image FSL eyes software, the atlas of 3T MR brain and a previous report of CRP of DTI study [15,23]. Two radiologists specializing in neuroimaging determined the ROI by referring to this, and the final ROI was agreed upon (Figure 1).

The FA values for each ROI were obtained by averaging all voxels within the ROI on the ipsilesional or contralesional sides with reference to the infarct. In each patient, FA of the CRP and CST was derived from the mean value of 15 contiguous sections. Additionally, the FA of CRP was reanalyzed by dividing it into three parts—lower pons, pontomedullary junction, and upper medulla—according to the anatomical levels on RF ROI of the brainstem. The FA values were obtained as the average of three contiguous sections per level, and the interval between levels were three contiguous sections. (Figure 2) Additionally, the ratio of the FA (rFA) between the ipsilesional and contralesional sides was calculated (rFA = FA ipsilesional side/FA contralesional side).

### 2.7. Statistical Analysis

All values were presented as the mean ± standard deviation. Ipsilesional values and contralesional values were compared using the paired *t*-test. Group comparisons were made using the independent *t*-test. All data from the complete set of assessments were analyzed using SPSS software version 25.0 (SPSS, Inc.; Chicago, IL, USA). A *p* value of 0.05 was used to indicate statistically significant differences.

## 3. Results

### 3.1. General Characteristics

In total, 70 patients were included in this study. Of these, 43 (61.4%) were men, and the mean age was 61.5 ± 11.5 years. Left lesions were noted in 31 (44.3%) patients. The middle cerebral artery was the most common lesion location, and the average score on the initial National Institute of Health Stroke Scale (NIHSS) was 6.8 ± 5.6. The patients underwent MR imaging at 27.1 ± 7.0 days after stroke. The two groups were divided according to the presence of spasticity. The control group contained 41 patients, and the spasticity group contained 29 patients. There were no significant intergroup differences in age, sex, infarction side, lesion, or interval between onset and MR DTI (*p* > 0.05). The NIHSS score was lower in the spasticity group (*p* = 0.03). The general characteristics of the subjects and each subgroup are shown in Table 1.

### 3.2. TBSS

TBSS revealed a significant decrease in FA values in several brain regions in the spasticity group in comparison with the control group for both lesions (pFWE < 0.05). These regions included the ipsilesional sensorimotor cortex, subcortical white matter, including the internal capsule and thalamic radiation, and brainstem in left ischemic stroke, and the ipsilesional brain stem in right ischemic stroke (Figure 3).

### 3.3. ROI Analysis of FA of the CST in the Brain Stem

The iFA values of the CST in the brain stem were significantly lower than the cFA values in both control and spasticity groups (*p* < 0.01). In comparison to the FA value of the control group, the FA value of the spasticity group was significantly lower in the ROIs of the CST in the ipsilesional brain stem (*p* < 0.01). There was no difference in the FA value in the ROIs of the CST in the contralesional brain stem. The rFA of the CST showed a statistically significant decrease in the spasticity group compared to the control group (*p* <0.01). (Table 2).

### 3.4. ROI Analysis of FA of the CRP in the Brain Stem

The iFAs of the CRP in the brain stem, which were the means of values recorded from the lower pontine RF to upper medullary RF, were significantly lower than the cFAs in the control group (*p* = 0.02). There was no difference between iFAs and cFAs in the spasticity group (*p* = 0.92). Compared to the control group, the iFA value in the spasticity group was not significantly different (*p* = 0.27), but the cFA value in the spasticity group was significantly lower (*p* = 0.04). There was no statistically significant difference in the rFA values of the CRP between the two groups (*p* < 0.01) (Table 2).

### 3.5. ROI Analysis of FA of the CRP in the Pons, Pontomedullary Junction, and Medulla

Table 3 shows the FA values obtained with sub-ROI analysis according to the anatomical structures in the CRP. In the ROIs of pontine RF, the iFA values of the CRP were significantly lower than the cFA values in both control and spasticity groups (*p* < 0.01). There were no differences in the iFA, cFA, or rFA values of the ROIs of the CRP in the pontine RF between the control and spasticity groups (*p* = 0.39, *p* = 0.42, and *p* = 0.84, respectively).

In the ROIs of the pontomedullary junction of RF, the iFA values and the cFA values of the CRP showed no differences in control and spasticity groups (*p* = 0.17 and *p* = 0.98). There were no differences in the iFA, cFA, or rFA values of the ROIs in the pontomedullary junction between control and spasticity groups (*p* = 0.24, *p* = 0.05, and *p* = 0.39, respectively).

In the ROIs of medullary RF, there was no difference between the iFA value and the cFA value in the control group (*p* = 0.33). However, the cFA value was lower than the iFA values of the CRP in the spasticity group (*p* < 0.01). There was no difference in the iFA between control and spasticity groups (*p* = 0.69). The cFA value of medullary RF in the spasticity group was significantly lower than that in the control group (*p* = 0.03). The rFA showed a significant difference between the control and spasticity groups (*p* = 0.02).

## 4. Discussion

We evaluated the relationship of CRP to PSS using DTI. In this study, we first used TBSS to confirm that secondary degeneration occurs in the brainstem in the subacute phase of supratentorial stroke. On the basis of these findings, we analyzed iFA and cFA values of ROIs of CST and CRP in the brainstem after supratentorial stroke and compared them within groups and between the control and spasticity groups.

In the DTI analysis of CST, the iFA values were lower than cFA values in both groups, and the iFA and rFA values in the spasticity group were lower than those in the control group. The total ROI analysis of the CRP from the lower pons to the upper medulla showed inconsistent results. In the control group, the iFA value of CRP was lower than the cFA value, and rFA values indicated ipsilesional/contralesional asymmetry. However, the spasticity group showed no asymmetry of FA values. Unlike the CST, CRP showed a significant difference in the cFA values between control and spasticity groups. In sub-ROI analysis of CRP, the cFA value in the spasticity group was lower than that in the control group at the medullary RF. Sub-ROI analysis of CRP suggested that injuries of ipsilesional CRP in supratentorial stroke caused degeneration of contralesional medullary RF, contributing to PSS.

The pathophysiology of PSS after stroke is still unclear. One of the pathophysiologies of PSS is the imbalance of dorsal RST and medial RST, and it is caused by injury to the CRP after stroke [6,8,10]. In our study, we analyzed the sub-ROIs of CRP by dividing it into the pons, pontomedullary junction, and medulla levels. This was based on the origination of the dorsal and medial RST, tracts that are mentioned to be important to the pathophysiology of PSS. Our results prove the hypotheses for the laterality dominance of CRP and the role of CRP in PSS. The iFA value was lower at the pontine RF, and the cFA value was lower at medullary RF. In other words, the injured CRP projected dominantly to the ipsilateral pontine RF, the origin of medial RST, and it also projected dominantly to contralesional medullary RF, the origin of dorsal RST. These results support the laterality dominance of CRP. Also, the FA values of the medullary RF showed a significant difference between the control and spasticity groups. It is assumed that the CRP injury causing downregulation of contralesional dorsal RST is related to PSS.

Among the major descending pathways of the human motor system, isolated CST lesions only produce weakness, loss of dexterity, hypotonia, and hyporeflexia [24,25,26]. Injuries of the CST are known to be not directly related to spasticity [27]. However, severe motor weakness is a risk factor for PSS [5]. In our study, ipsilesional CST injury was confirmed using DTI after hemispheric stroke with motor weakness, and it was more severe on the spastic side after stroke. Worse CST injury is found in stroke patients with spasticity; this only explains the motor weakness of patients with spasticity. In other words, the motor weakness with spasticity in stroke patients is caused by having a projection adjacent between CST and CRP.

MR imaging studies of stroke have been actively conducted. Previous studies reported significant correlations between DTI biomarkers such as FA and motor impairments, and the DTI biomarkers measured after stroke have emerged as potential predictors of motor recovery [11]. Most studies on motor impairment after stroke focused on motor weakness with CST injury. However, MR imaging of PSS is limited to studies of stroke volume and lesions. They reported that extensive lesion involvement of more than one cerebral lobe was frequent in patients with PSS and injury to PMC, putamen, internal capsule posterior limb, external capsule, thalamus, and insula were correlated with PSS [28,29,30,31]. Among these, a recent study reported that PMC was related to PSS, and it was attributed to the origin of CRP [29]. Moreover, most of the above-mentioned subcortical lesions were related to the projections of the CRP in the subcortical white matter as well as the CST, and it reflected that the CRP contributed to the pathophysiology of PSS.

With the development of MR image analysis, DTI analyses of pathways other than CST are also being attempted. Jang and colleagues have implemented diffusion tensor tractography of CRP in the human brain and published several studies related to the functional neuroanatomy of the CRP [9,15,16]. They showed that the first cortical origin area of the CRP was the PMC, and the CRP was located close anteromedially to the CST in subcortical white matter. They also reported many studies on the role and recovery of CRP in walking ability in relation to posture and locomotion, the major roles of RST [17,18,19,20,32,33,34]. Although the DTI studies of CRP related to walking ability or balance after stroke and the reports on the role of CRP in the pathophysiology of PSS have increased, there were no studies on DTI of CRP in PSS.

This is the first study to evaluate the relationship between the CRP and PSS by using DTI. This study revealed that CRP injury was related to the spasticity caused by hemispheric stroke. It also suggested that injury to the CRP caused degeneration in the contralesional medullar RF, which is the origin of the dorsal RST, resulting in an imbalance of RST, one of the pathophysiologies of PSS. Additionally, the results suggest that CRP injury could lead to secondary degeneration in the brain stem in subacute strokes as well as CST injuries.

This study did have some limitations. First, it was a retrospective study, so selection bias could have occurred. Second, when acquiring MR images for DTI analysis, the voxel size should be isotopic, but our images are not, so there is a possibility of the inclusion of minor errors. Also, the patients with lesions in the ROIs, with cerebral edema, or with hemorrhagic transformation which had possible structural displacements of brain tissue were excluded from the study; however, the effect of the structural lesion by stroke cannot be completely ruled out. Third, our subject was an elderly patient, and it is highly likely that they have small vascular lesions, so the effect of small vascular lesions could not be excluded. Fourth, due to the limitations of MR DTI techniques, among the imbalance of inhibitory and excitatory effects to spinal reflex circuits, which is the strongest pathophysiology of spasticity, our results only showed an inhibitory input of the dorsal RST. The upregulation of medial RST, an excitatory input, could not be estimated. Further imaging studies performed with advanced imaging techniques using fMRI and/or electrophysiological evaluation using transcranial magnetic stimulation may provide direct evidence of the pathogenesis of PSS in the future.

## 5. Conclusions

We investigated whether CRP injuries through supratentorial lesions were associated with PSS using DTI. The decreased FA values in the contralesional medullary RF of CRP seems to be related to the degeneration of dorsal RST, which is the dominant inhibitory effect of the spinal stretch reflex, in PSS. Therefore, these findings could provide imaging evidence of the pathophysiology of PSS, which is the imbalance between descending inhibitory and facilitatory regulation of spinal stretch reflexes after stroke.

## Figures and Tables

**Figure 1 jpm-11-01151-f001:**
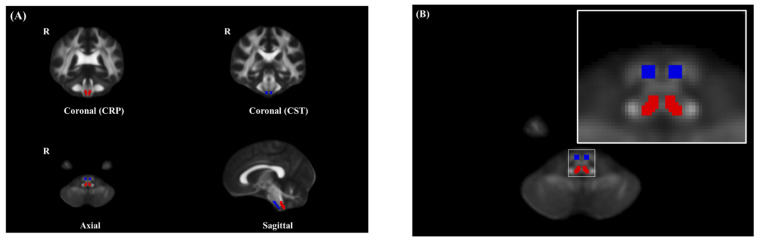
(**A**) ROIs of the left and right CRP and CST at the level of the lower pons to the upper medulla based on the T2-weighted image (Red color, CRP; Blue color, CST). (**B**) ROIs of the left and right CRP and CST at the medulla. ROI, region of interest; CST, corticospinal tract; CRP, corticoreticular pathway.

**Figure 2 jpm-11-01151-f002:**
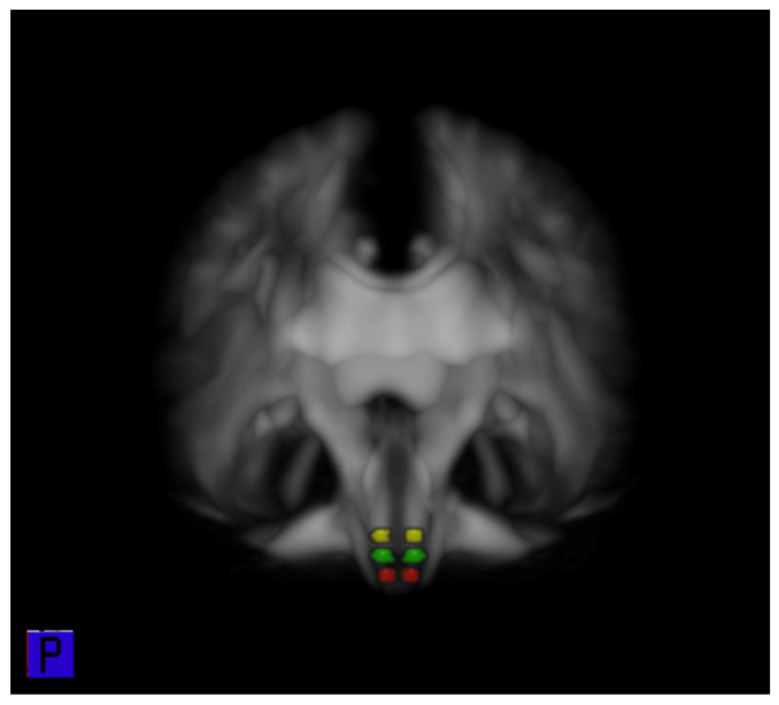
Sub-ROIs of the left and right CRP at RF on the brain stem (Yellow color; pontine RF, Green color; pontomedullary junction RF, Red color; medullar RF). ROI, region of interest; CRP, corticoreticular pathway; RF, reticular formation.

**Figure 3 jpm-11-01151-f003:**
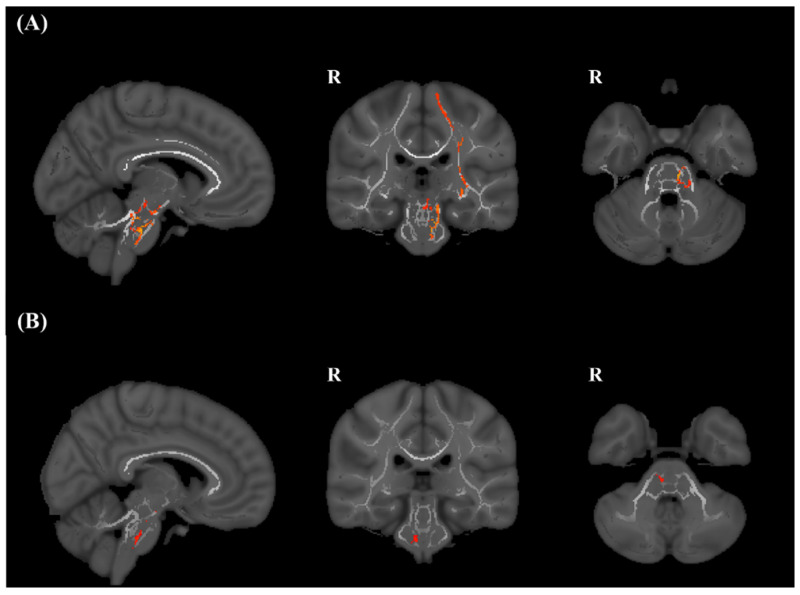
TBSS analysis between control and spasticity groups. (**A**) TBSS of the left hemispheric stroke, (**B**) TBSS of the right hemispheric stroke. There was a significant decrease in fractional anisotropy values in the ipsilesional brain stem in the spasticity group, compared with the control group in both lesions (*p* < 0.05). TBSS, Tract-based spatial statistics.

**Table 1 jpm-11-01151-t001:** Demographic and clinical data of participants.

	Total (*n* = 70)	Control (*n* = 41)	Spasticity (*n* = 29)	*p*
Age, mean ± SD (years)	61.6 ± 11.5	61.7 ± 12.3	61.5 ± 10.3	0.934
Sex				0.932
Male, *n* (%)	43 (61.4%)	25 (61.0%)	18 (62.1%)	
Female, *n* (%)	27 (38.6%)	16 (39.0%)	11 (37.9%)	
Infarction side				0.939
Lt., *n* (%)	31 (44.3%)	18 (43.9%)	13 (44.8%)	
Rt., *n* (%)	39 (55.7%)	23 (56.1%)	16 (55.2%)	
Lesion, *n*				0.530
MCA	52 (74.3%)	29 (70.7%)	23 (79.3%)	
LSA	10 (14.3%)	7 (17.1%)	3 (10.3%)	
Multiple infarction	8 (11.4%)	5 (12.2%)	3 (10.3%)	
NIHSS score, mean	6.8 ± 5.6	5.9 ± 4.9	8.6 ± 6.1	0.032 *
MR DTI after onset, mean ± SD (days)	27.1 ± 7.0	26.8 ± 7.9	27.6 ± 5.7	0.679
MAS, *n* (%)				<0.001 *
0	41 (58.6%)	41 (100%)	-	
1	10 (14.3%)	-	10 (34.5%)	
1+	12 (17.1%)	-	12 (41.4%)	
2	5 (7.1%)	-	5 (17.2%)	
3	2 (2.9%)	-	2 (6.9%)	
4	0 (0.0%)	-	0 (0.0%)	

* *p* < 0.05; SD, Standard deviation; MCA, Middle cerebral artery; LSA, Lenticulostriate artery; NIHSS, National Institute of Health Stroke Scale; DTI, diffusion tensor image; MAS, modified Ashworth scale.

**Table 2 jpm-11-01151-t002:** Comparison of the FA values of CRP and CST in the brainstem between the control and spasticity groups.

ROIs of FA		Control	Spasticity	*p*
CST	iFA	0.371 ± 0.547	0.308 ± 0.529	<0.01 *
	cFA	0.396 ± 0.514	0.390 ± 0.689	0.91
	*p*	<0.01 *	<0.01 *	
	rFA	0.952 ± 0.106	0.802 ± 0.147	<0.01 *
CRP	iFA	0.403 ± 0.026	0.394 ± 0.036	0.27
	cFA	0.412 ± 0.035	0.395 ± 0.032	0.04 *
	*p*	0.02 *	0.92	
	rFA	0.981 ± 0.056	1.000 ± 0.053	0.18

* *p* < 0.05, All values are presented as the mean ± standard deviation. CST, corticospinal tract; CRP, corticoreticular pathway; ROI, region of interest; iFA, ipsilesional fractional anisotropy; cFA, contralesional fractional anisotropy; rFA, ratio between ipsilesional/contralesional fractional anisotropy.

**Table 3 jpm-11-01151-t003:** Comparison of the FA values on sub-ROIs of CRP in the control and spasticity groups.

ROIs of FA		Control	Spasticity	*p*
Pontine RF	iFA	0.512 ± 0.046	0.501 ± 0.057	0.39
	cFA	0.526 ± 0.057	0.515 ± 0.052	0.42
	*p*	0.01 *	0.03 *	
	rFA	0.978 ± 0.062	0.975 ± 0.063	0.84
Pontomedullar junction RF	iFA	0.364 ± 0.031	0.354 ± 0.037	0.24
	cFA	0.372 ± 0.036	0.354 ± 0.036	0.05
	*p*	0.17	0.98	
	rFA	0.985 ± 0.089	1.003 ± 0.084	0.39
Medullar RF	iFA	0.346 ± 0.029	0.343 ± 0.045	0.69
	cFA	0.350 ± 0.035	0.330 ± 0.039	0.03 *
	*p*	0.33	0.02 *	
	rFA	0.992 ± 0.073	1.040 ± 0.085	0.02 *

* *p* < 0.05, All values were presented are the mean ± standard deviation. ROI, region of interest; RF, reticular formation; iFA, ipsilesional fractional anisotropy; cFA, contralesional fractional anisotropy; rFA, ratio between ipsilesional/contralesional fractional anisotropy.

## Data Availability

The data that support the findings of this study are available from the corresponding author upon reasonable request.

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
