# Peer review of "Corticoreticular Pathway in Post-Stroke Spasticity: A Diffusion Tensor Imaging Study"

_jpm, 2021, doi:10.3390/jpm11111151_

Round 1

Reviewer 1 Report

The topic of the paper is important as it could provide imaging evidence of the pathophysiology of post-stroke spasticity. However several revisions are needed.

Abstract

  1. Add an introduction phrase in the abstract, including background and not to begin with the aim of the study.

Main text

  1. the previous literature on the correlation between neuroimaging and the development of post stroke spasticity is greater than what the authors reported. I suggest citing a greater number of related works, having omitted some with a high sample size; among these for example: Cheung et al. Lesion Characteristics of Individuals With Upper Limb Spasticity After Stroke.
  2. the authors point to their work as the first assessing the CRP in post-stroke spasticity by using DTI. However, I believe it is usefully to refer to the recently published paper by SoYeon Jun et al (“Does Motor Tract Integrity at 1 Month Predict Gait and Balance Outcomes at 6 Months in Stroke Patients?”), although the target is mainly on walking.
  3. It would be interesting to integrate information on the type of rehabilitation treatment carried out, as this could also influence the clinical outcome and the degree of spasticity.

Minor review

1. I would not use the acronym UMN, as it was used only once in the following text

2. Line 73. Please verify “Yansan Hospital”

Author Response

Response to Reviewer 1 Comments

The topic of the paper is important as it could provide imaging evidence of the pathophysiology of post-stroke spasticity. However several revisions are needed.

Thank you for your opinion on our article, which had some shortcomings. We have made the revisions to reflect your comments.

Abstract

  1. Add an introduction phrase in the abstract, including background and not to begin with the aim of the study.

Response) Thanks for your comment. It was difficult to summarize our study in less than 200 words in the abstract. We added the introduction to the beginning of the abstract.

[Line 13-15]

“One of the pathophysiologies of post-stroke spasticity (PSS) is the imbalance of the reticulospinal tract (RST) caused by injury to the corticoreticular pathway (CRP) after stroke.”

According to the opinions of another reviewer, we also have revised the abstract overall to help readers understand our study more clearly.

:

Main text

  1. the previous literature on the correlation between neuroimaging and the development of post stroke spasticity is greater than what the authors reported. I suggest citing a greater number of related works, having omitted some with a high sample size; among these for example: Cheung et al. Lesion Characteristics of Individuals With Upper Limb Spasticity After Stroke.

Response) Thanks for suggesting the important article (Cheung et al. Lesion Characteristics of Individuals With Upper Limb Spasticity After Stroke.) that we missed. We have additionally cited the article you recommended. Most of the lesions presented in the report were included in our previous manuscript, so only the insula was added.

[Line 284-287]

They reported that extensive lesion involvement of more than one cerebral lobe was fre-quent in patients with PSS and injury to PMC, putamen, internal capsule posterior limb, external capsule, thalamus, and insula were correlated with PSS [28-31]

  1. the authors point to their work as the first assessing the CRP in post-stroke spasticity by using DTI. However, I believe it is usefully to refer to the recently published paper by SoYeon Jun et al (“Does Motor Tract Integrity at 1 Month Predict Gait and Balance Outcomes at 6 Months in Stroke Patients?”), although the target is mainly on walking.

Response) Thank you very much for your comments about recent research that we did not check.

In the introduction, I mentioned that there are studies on the relationship between CRP and the gait function.

 “Since then, studies on CRP injury after stroke and traumatic brain injury have also been conducted using DTI. However, DTI studies of CRP after stroke were few and only contain important information about the relationship between CRP and gait function [17-20].”

However, the article you mentioned is missing, so I added the citation. It was also included in the dicussion.

In addition, previous studies, including the study you recommended, tried to confirm the association between CRP and balance or gait, but the relationship between CRP and spasticity was not assessed at the studies, so we argued that it was the first report. To avoid misunderstanding, we have made the following changes.

[Line 302]

“This is the first study assessing the CRP in post-stroke spasticity by using DTI”

  • “This is the first study to evaluate the relationship the CRP and PSS by using DTI.”

  1. It would be interesting to integrate information on the type of rehabilitation treatment carried out, as this could also influence the clinical outcome and the degree of spasticity.

Response) All subjects were in patients after stroke and were receiving inpatient rehabilitation after stroke according to the clinical pathway of our center. There were no subjects who did not receive rehabilitation treatment. Rehabilitation treatment was given as intensity as possible depending on the patient's condition. If it is recommended to add information about the subject's rehabilitation intensity, we will add it.

[Line 92-93]

All subjects underwent conventional inpatient rehabilitation including physical therapy and occupational therapy after stroke in our center.

Minor review

  1. I would not use the acronym UMN, as it was used only once in the following text

Response) Thank you for your kind review for checking for unnecessary acronyms. We changed UMN to 'upper motor neuron'.

  1. Line 73. Please verify “Yansan Hospital”

Response) Thanks for checking the typographical error. We corrected it to “Yangsan Hospital”

Reviewer 2 Report

This is the retrospective study that investigate the pathophysiology of post-stroke spasticity using DTI. The idea seems interesting, but study has several limitations that may influence the results. In general, the manuscript could be more concise. Abstract seems to be chaotic and makes the main idea of the study and main finding difficult to understand. I included further comments below.   

Abstract:

In my opinion abstract is not informative, some abbreviations are not explained (RST). It is hard to understand what authors actually showed in their study.

Introduction:

The authors should more clearly explain the role of CRP and RST. Also, short summary of what exactly was found in previously published studies should be provided. The aim of the study is very general, and introduction does not provide enough background to understand it.

Methods:

What is the rationale behind exclusion patients older than 80yo?

In exclusion criteria – what authors understand by “any previous brain lesion”. Were all patients with small vascular lesions excluded?

Brain lesion after stroke can disrupt linear and non-linear registration to MNI space. How did authors deal with that problem? Were the lesions masked before registration?

Results:

I am concerned about reproducibility of ROI placement. Where there any landmark used? Were FA from ROI calculated in MNI space or subject space?

In Table 1 the p-value for comparison between control and spasticity could provide additional information on potential difference between two groups.

In Figure 3 sides (R/L) could be indicated on the pictures to facilitate interpretation.

How did authors deal with problem of multiple comparison?

Discussion:

It should be clearly stated what are the main findings of the study.

Discussion in general should be more concise.

In limitation section the possible impact of stroke lesion on image processing should be discussed. Were macroscopically seen stroke lesion excluded from the analysis? What about area of stroke? What about multiple comparisons?

Author Response

Response to Reviewer 2 Comments

This is the retrospective study that investigate the pathophysiology of post-stroke spasticity using DTI. The idea seems interesting, but study has several limitations that may influence the results. In general, the manuscript could be more concise. Abstract seems to be chaotic and makes the main idea of the study and main finding difficult to understand. I included further comments below.

Thank you for your opinion on our article, which had some shortcomings. We have made the revisions to reflect your comments.

Abstract:

  1. In my opinion abstract is not informative, some abbreviations are not explained (RST). It is hard to understand what authors actually showed in their study.

Response) Thank you so much. We acknowledge our shortcomings and agree with your opinion. According to your opinions, we have revised the abstract overall to help readers understand our study more clearly. We also checked abbreviations. However it is still difficult to summarize our study in less than 200 words in the abstract. If there are still some shortcomings in the revised abstract, please let us know. The revised abstract is 195 words.

Introduction:

  1. The authors should more clearly explain the role of CRP and RST. Also, short summary of what exactly was found in previously published studies should be provided. The aim of the study is very general, and introduction does not provide enough background to understand it.

Response) Thank you very much. According to your comments, we moved some contents of CRP and RST in the discussion to the introduction, so we tried to describe the pathophysiology of PSS related CRP and RST after stroke, which provided in the previous studies.

Methods:

  1. What is the rationale behind exclusion patients older than 80yo?

Response) There was no special rationale for this. However, we tried to reduce the effects of a mini-stroke, white matter change-related aging, or small vessel lesion by limiting the super-aged.

  1. In exclusion criteria – what authors understand by “any previous brain lesion”. Were all patients with small vascular lesions excluded?

Response) Small vascular lesions were not excluded. Patients with definite anatomical brain lesions such as previous stroke, TBI, brain tumor, etc. were excluded. We rewrote as “any previous brain lesion such as previous stroke, traumatic brain injury, brain tumor, etc.

We don't know if your intention is correct, but we thought it was necessary to mention the effect that small vascular lesions can have, so we wrote it in the limitation.

[Line 311-312]

“Third, our subject is an elderly patient, and it is highly likely that they have small vascular lesions, so the effect of small vascular lesions could not be excluded.”

  1. Brain lesion after stroke can disrupt linear and non-linear registration to MNI space. How did authors deal with that problem? Were the lesions masked before registration?

Response) Thanks for your comment.

As you mentioned, abnormal brain deformation by stroke can cause the misregistration of common space. DTI was spatially normalized to the MNI template using parameters derived from anatomical T1WI processing. Brain shape for stroke patients was globally intact; it means that linear registration step is not problematic, but the regions nearby infarct and/or hemorrhage site may be locally deformed in subsequent nonlinear registration steps. However, we measured the DTI index in a manually drawn ROI area spatially far away from the stroke damage sites. Therefore, we add to explain about using parameters derived from anatomical T1WI processing.

[Line 107-114]

All subjects underwent 3.0T MRI scanner (Skyra, Siemens Healthneers, Germany) equipped with a 16-channel head & neck coil to acquire 3D T1-weighted and DTI images. The 3D T1-weighted images were obtained using magnetization prepared rapid acquisition gradient recalled echo (MPRAGE) pulse sequence with the following parameters: repetition time (TR) / echo time (TE) / inversion time (TI) = 1900 / 2.2 / 900 ms, Flip angle (FA) = 9º, 1mm3 isotropic voxel size. DTI data were acquired at an average of 32.1 ± 7.0 days after stroke onset by conventional brain MRI protocols using the echo-planar imaging sequence.

[Line 122-128]

All DTI data was spatially normalized to Montreal Neurological Institute (MNI) template using parameters derived from 3D T1WI processing. DTI analysis was performed with FMRIB's Diffusion Toolbox and TBSS (Tract-Based Spatial Statistics) in the FMRIB Software Library 6.0 (FSL, https://fsl.fmrib.ox.ac.uk/fsl/fslwiki/FSL) package. For registration, was performed with FMRIB Software Library 6.0 and get the transformation matrix with FMRIB’s linear registration tool(FLIRT) and FMRIB’s nonlinear registration tool(FNIRT).

Results:

  1. I am concerned about reproducibility of ROI placement. Where there any landmark used? Were FA from ROI calculated in MNI space or subject space?

Response)

Thank you. We referred to the following paper of corticoreticular tract of diffusion tensor tractography study as a reference 18 and we cited it. Also, the reference 26 is atlas of 3T MR brain. The book showed magnetic resonance images side-by-side with corresponding brain anatomy of the reticular formation. Two radiologists specializing in neuroimaging determined the ROI by referring to this, and the final ROI was agreed upon. Also, we obtained FA values in MNI space.

We added the following sentences:

[Line 122-123]

All DTI data was spatially normalized to Montreal Neurological Institute (MNI) template using parameters derived from 3D T1WI processing.

[Line 145-147]

…atlas of 3T MR brain and a previous report of CRP of DTI study [18,26]. Two radiologists specializing in neuroimaging determined the ROI by referring to this, and the final ROI was agreed upon (Fig. 1).

  1. In Table 1 the p-value for comparison between control and spasticity could provide additional information on potential difference between two groups.

Response) Added below based on your comments.

Table 1. Demographic and clinical data of participants.

Total

(n.=70)

Control

(n.=41)

Spasticity

(n.=29)

p

Age, mean±SD (years)

61.6±11.5

61.7±12.3

61.5±10.3

0.934

Sex

0.932

Male, n. (%)

43 (61.4%)

25 (61.0%)

18 (62.1%)

Female, n. (%)

27 (38.6%)

16 (39.0%)

11 (37.9%)

Infarction side

0.939

Lt., n (%)

31 (44.3%)

18 (43.9%)

13 (44.8%)

Rt., n (%)

39 (55.7%)

23 (56.1%)

16 (55.2%)

Lesion, n.

0.530

MCA

52 (74.3%)

29 (70.7%)

23 (79.3%)

LSA

10 (14.3%)

7 (17.1%)

3 (10.3%)

Multiple infarction

8 (11.4%)

5 (12.2%)

3 (10.3%)

NIHSS score, mean

6.8±5.6

5.9±4.9

8.6±6.1

0.032*

MR DTI after onset, mean±SD (days)

27.1±7.0

26.8±7.9

27.6±5.7

0.679

MAS, n (%)

<0.001*

0

41 (58.6%)

41 (100%)

-

1

10 (14.3%)

-

10 (34.5%)

1+

12 (17.1%)

-

12 (41.4%)

2

5 (7.1%)

-

5 (17.2%)

3

2 (2.9%)

-

2 (6.9%)

4

0 (0.0%)

-

0 (0.0%)

*p<0.05; SD, Standard deviation; MCA, Middle cerebral artery; LSA, Lenticulostriate artery; NIHSS, National Institute of Health Stroke Scale; DTI, diffusion tensor image; MAS, modified Ashworth scale.

  1. In Figure 3 sides (R/L) could be indicated on the pictures to facilitate interpretation.

Response) Thank you so much. Add on the picture. Please see the attachment.

  1. How did authors deal with problem of multiple comparison?

Response) We are very sorry to you. We didn't understand what multiple comparison meant. If you give me a little more detail, we will reply again.

Discussion:

  1. It should be clearly stated what are the main findings of the study. Discussion in general should be more concise.

Response) I agree with your opinion. As the discussion was long, the results of our study are blurred, and it seems that unnecessary content is included. 1) The explanation of the CST results was thought to be unnecessarily long, so it was greatly shortened. 2) some of the contents of the pathophysiology of PSS, including CRP and RST, were moved to the introduction, and only mentioned in the discussion.

Discussion has been reordered and modified as a whole. Many sentences were deleted. We didn't remark the sentences we deleted. Thank you for your opinion.

  1. In limitation section the possible impact of stroke lesion on image processing should be discussed. Were macroscopically seen stroke lesion excluded from the analysis? What about area of stroke? What about multiple comparisons?

Response) Most of the answers to the impact of stroke lesion on image processing are described in method 2 and 3. except multiple comparison

In addition to that, patients with brainstem lesion corresponding to ROI were also excluded (exclusion criteria (2)). We also excluded patients with hemorrhagic transformation or cerebral/cerebellar edema after cerebral infarction (exclusion criteria (3)).

Nevertheless, if there is something you think is a limitation, please do not hesitate to suggest it to us.

Round 2

Reviewer 1 Report

I congratulate with the authors for the modification that improved the quality of the paper. I don't have any other suggestion.

Author Response

Thank you for your advice. My report has been upgraded through your comments. Thanks again.

Reviewer 2 Report

I appreciate the authors effort to address my suggestions.  The manuscript seems to be much improved now. However, I still have methodological concerns regarding brain lesions and pre-processing steps.

The issue of brain lesion is still not clear to me.

--Authors included stroke patients with mean NIHSS score around 6.8±5.6 which is a moderate severity and performed MRI scan 3-6 weeks after stroke. It is very unlikely that none of them had structural brain lesion that need to be masked in non-linear registration in FSL.

--As main result refers to ROI subanalysis it should be clearly describe in methods how the ROI were placed. Were the FA value calculated in native or MNI space? Were ROI manually placed in native or common MNI space?  The coordinates or landmarks for ROI selection should be provided.
